# The Delivery of the Novel Drug ‘Halicin’ Using Electrospun Fibers for the Treatment of Pressure Ulcer against Pathogenic Bacteria

**DOI:** 10.3390/pharmaceutics12121189

**Published:** 2020-12-08

**Authors:** Walaa S. Aburayan, Rayan Y. Booq, Nouf S. BinSaleh, Haya A. Alfassam, Abrar A. Bakr, Haitham A. Bukhary, Essam J. Alyamani, Essam A. Tawfik

**Affiliations:** 1National Center for Pharmaceutical Technology, King Abdulaziz City for Science and Technology (KACST), P.O. Box 6086, Riyadh 11442, Saudi Arabia; waburayan@kacst.edu.sa (W.S.A.); nbinsaleh@kacst.edu.sa (N.S.B.); 2National Center of Biotechnology, King Abdulaziz City for Science and Technology (KACST), P.O. Box 6086, Riyadh 11442, Saudi Arabia; rbooq@kacst.edu.sa (R.Y.B.); aabakr@kacst.edu.sa (A.A.B.); 3Center of Excellence for Biomedicine, King Abdulaziz City for Science and Technology (KACST), P.O. Box 6086, Riyadh 11442, Saudi Arabia; halfassam@kacst.edu.sa; 4Department of Pharmaceutics, College of Pharmacy, Umm Al-Qura University, Makkah 24381, Saudi Arabia; habukhary@uqu.edu.sa

**Keywords:** electrospinning, electrospun fibers, halicin, antibacterial dressing, pressure ulcer

## Abstract

Pressure ulcer or bedsore is a form of skin infection that commonly occurs with patients admitted to the hospital for an extended period of time, which might lead to severe complications in the absence of medical attention, resulting in infection either by drug-sensitive or drug-resistant bacteria. Halicin, a newly discovered drug effective against several bacterial strains, including multidrug-resistant bacteria, was investigated to reduce bacterial infection burden. This study aims to formulate halicin into electrospun fibers to be applied in bedsores as antibacterial dressing to assess its efficacy against gram-positive (*Staphylococcus aureus*) and gram-negative bacteria (*Escherichia coli* and *Acinetobacter baumannii*) by studying the minimum inhibitory concentration (MIC) and bacterial zone of inhibition assays. The diameters of inhibition growth zones were measured, and the results have shown that the drug-loaded fibers were able to inhibit the growth of bacteria compared to the halicin discs. The release profile of the drug-loaded fibers exhibited a complete release of the drug after 2 h. The results demonstrated that the drug-loaded fibers could successfully release the drug while retaining their biological activity and they may be used as a potential antimicrobial dressing for patients with pressure ulcers caused by multidrug resistant bacteria.

## 1. Introduction

Pressure ulcers (also known as bedsore) is a form of infection that breaks down the skin and the underlying tissues causing tissue ischemia and necrosis [1]. Normal skin tissues can hold a pressure of about 30–32 mm Hg on the arterial side for just a brief period [2,3,4]. Nevertheless, as the pressure rises slightly above, it induces microcirculatory occlusion, resulting in ischemia (i.e., tissue death) and ulceration [3,4]. The tissue distortion happens when the soft tissues are squeezed or sheared between the skin while sitting or standing [5]. In addition to occluding blood flow, tissue obstruction often hinders the lymphatic drainage, which may contribute to the aggregation of metabolic waste materials, proteins, and enzymes in the infected tissues, leading to tissue injury [6,7,8]. The majority of people impacted by pressure sores are admitted to the hospital for an extended time. Several different health conditions affect blood flow and capillary perfusion, such as type 2 diabetes, making the person more susceptible to pressure ulcers [9].

Electrospinning is a versatile process for fabricating fibers with diameters that can reach nanoscale. Owing to the large surface area, ease of preparation, flexibility, and tensile strength of these fibers, they have been utilized in a wide range of applications, including antimicrobial dressings tissue engineering, wound healing, drug delivery, and in the manufacture of textiles, cosmetics, molecular filtration, fuel cells, and sensor systems [10,11]. Besides, the ability to encapsulate different active compounds, such as drugs, peptides, proteins, nucleic acids, and cells, makes these fibers an attractive drug delivery platform. The diversity of the natural and synthetic occurring polymers allows the fabrication of fibers that are compatible with the biological tissues, which can hold a potential significance in tissue regeneration applications. The resemblance of the fibers to the elements of tissue extracellular matrix (ECM), which plays an essential role in wound healing, allows the use of these fibers as scaffolds to accelerate wound closure [11].

To treat these skin ulcers, advanced wound dressings have gained considerable interest, as they can protect the wound from infection, absorb exudates during wound moisturization, and accelerate wound healing. Owing to their vast surface area and porous nature, electrospun fibers display a great potential as wound dressing, which can absorb exudates, in addition to their ability to be loaded with medicinal agents, such as drugs and growth factors [12,13,14,15]. Various studies have shown that electrospun wound dressings play efficient roles in hemostasis, antibacterial effects, scar formation reduction, and skin regeneration [16,17]. The skin’s barrier-like nature can be a challenge to deliver some hydrophilic drugs, whereas it is possible to deliver small to low molecular weight (up to a few hundred Daltons) and hydrophobic medicines [18,19]. Therefore, electrospun fibers are intended to facilitate the delivery of therapeutic agents into the body via the skin.

Numerous studies have explored herbal medicinal compounds, such as curcumin from turmeric [20], aloesin from aloe vera [21], or thymol from thyme [22], loaded into electrospun nanofibers as drug-loaded wound dressings [23]. In the case of pressure ulcers, hydrocolloid dressings [24,25], alginate dressings [26], and nanosilver dressings [27,28] are commonly used as wound dressings. However, in some cases, antibiotics are necessary and required to target different types of resistant bacteria. Therefore, discovering novel antibiotics with low-to-none bacterial resistance is essential to overcome the rise of antibiotic-resistant bacterial strains [29].

Among surgical, organ transplantation, cancer, and critical care patients, antibiotic therapies were able to enhance such population’s clinical outcomes throughout the reduction of infection-associated mortality. According to the World Health Organization (WHO) 2019 report, approximately 700,000 people die every year from antibiotic-resistant infections. In the United States, nearly two million are infected with antibiotic-resistant bacteria yearly, with a mortality of 23,000 of these cases according to Center for Disease Control and Prevention (CDC) [30]. MIT, Massachusetts Institute of Technology, scientists have recently discovered a promising new antibiotic compound using artificial intelligent (AI) technology [31]. In this study, a drug named ‘Halicin’ inhibited different gram-positive and gram-negative bacteria, including certain antibiotics-resistant strains. The researchers claimed that this study can be a model to discover, or even develop, new antimicrobial agents.

Halicin was intended to treat patients with diabetes [31,32,33]. However, due to AI technology, the researchers found that this drug could inhibit some bacterial strains, such as *Clostridium difficile, Acinetobacter baumannii,* and *Mycobacterium tuberculosis*, but not *Pseudomonas aeruginosa* because of the insufficient permeability to its cell membrane [34,35]. Halicin (5-[(5-Nitro-2-thiazolyl) thio]-1,3,4thiadiazol-2-amine) structure is more analogous to a class of anti-parasitic drugs that contains nitrogen and metronidazole antibiotic [31,36]. Additionally, this drug exhibited unique antibacterial mechanisms that involve the downregulations of bacterial genes responsible for their motility and the upregulation of iron homeostasis genes [31,37]. It can also interrupt the transmembrane electrochemical gradient leading to bacterial death. Therefore, this study aims to formulate a drug-loaded antibacterial dressing via electrospinning technique, which can potentially treat bedsores caused by multidrug-resistant bacteria.

## 2. Materials and Methods

### 2.1. Materials

PVP—polyvinylpyrrolidone, average Mw ~1,300,000, ethyl alcohol (≥99.5%) sodium chloride (≥99.5%), sodium phosphate dibasic (≥99.0%), potassium chloride (99.0–100.5%), and potassium phosphate monobasic (≥99.0%) were purchased from Sigma-Aldrich (St. Louis, MO, USA) Halicin (5-[(5-Nitro-2-thiazolyl) thio]-1,3,4thiadiazol-2-amine) was bought from Carbosynth Ltd. (Compton, UK). Mueller–Hinton broth was obtained from Scharlab, S.L (Barcelona, Spain). Deionized water (DW) was generated through Milli Q, Millipore (Billerica, MA, USA), and has been used throughout the study.

### 2.2. Preparation of Electrospinning Solution

PVP in a concentration of 8% (*w/v*) was dissolved in ethanol. The solution was stirred on a magnetic stirrer for 2 h at room temperature to achieve a clear homogenous solution. For the drug-loaded fibers, halicin with a concentration of 1% (*w/v*) was added to the PVP solution and stirred for an hour to obtain a homogenous solution. This drug concentration was chosen to achieve a polymer:drug ratio of 8:1, a recommended ratio that ensures a complete encapsulation of the drug within the fibrous matrix [38].

### 2.3. Preparation of Monoaxial Fibers

Monoaxial (i.e., single-layered) fibers were prepared using an electrospinning setup purchased from Spraybase^®^ electrospinning instrument (Spraybase^®^, Dublin 2, Ireland). The spinning solution was transferred to a 5 mL plastic syringe with an inner needle diameter of 0.9 mm. A syringe pump controlled the feed solution flow rate at a rate of 0.8 mL/h. The used voltage was between 6.5 and 8.5 kV for the blank and drug-loaded fibers, while the tip-to-collector distance was maintained at a distance of 15 cm. The metal collector was covered by aluminum foil, which is used as a substrate for deposition. The electrospinning process was carried out in the enclosed electrospinning apparatus at ambient temperature and relative humidity of about 30 to 35%.

### 2.4. Scanning Electron Microscopy (SEM)

The morphological characterization of the fibers was evaluated by the SEM (JSM-IT500HR SEM, JEOL Inc., Peabody, MA, USA). Fibers were collected directly onto the aluminum foil, and the samples were not coated with any conductive material. Fiber size analysis was performed by measuring the fibers’ diameter of at least 15 fibers using the microscope’s software (SEM Operation, 3.010, Akishima, Tokyo, Japan).

### 2.5. Fourier-Transform Infrared Spectroscopy (FTIR)

The physical state characterization was performed on the pure polymer and drug, their physical mixture (PM), and the blank and drug-loaded fibers and analyzed by Agilent Cary 630 ATR-FTIR analyzer (Agilent Technologies Inc.). Measurements were carried out at ambient temperature with a resolution of 4 cm^−1^ and a period of 4 scans/sample over a spectrum of 4000–650 cm^−1^. The obtained spectra were analyzed and plotted by OriginPro 2016 (OriginLab Corporation, Northampton, MA, USA).

### 2.6. X-Ray Diffraction (XRD)

Solid-state characterization was conducted using a MiniFlex 600 benchtop diffractometer (RigaKu, Tokyo, Japan). The samples were directly mounted on the glass trays and were studied at 40 kV and 15 mA, a Cu Kα radiation point source (λ = 1.5148 227 Å). XRD patterns were recorded from 2° to 50° (step size of 0.05° and time per step of 0.2 s) using a diffraction angle (2θ). Results were plotted and analyzed by OriginPro 2016 (OriginLab Corporation, Northampton, MA, USA).

### 2.7. Ultraviolet/Visible Analysis (UV/Vis)

A known concentration of the drug (i.e., 200 μg/mL) was dissolved in DW (for drug loading determination) or phosphate buffer saline (PBS) with a pH 7.4 (for release study) and subjected to UV/Vis scanning in the LAMBDA 850+ UV/Vis Spectrophotometer (PerkinElmer, Waltham, MA, USA) from 800 to 200 nm to obtain the absorption maximum (λ max) of the drug. After determining the drug λ max, the drug concentrations, ranging from 12.5 to 0.2 μg/mL, were measured at 340 nm (λ max), and standard curves were plotted against each concentration point’s absorbance.

### 2.8. Drug Loading (DL), Entrapment Efficiency (EE%), and Fiber Yield (Y)

To determine the *DL* and *EE*% of the drug-loaded fibers, three square pieces of the fibrous mat, with known weights of approximately 4 mg, were dissolved in DW and kept at ambient temperature for at least 4 h. The absorptions of these samples were recorded at 340 nm, and the *DL* was calculated according to the following equation:(1)$$DL=\frac{Entrapped\;drug\;amount}{Yield\;of\;fibres\;amount}$$

While the *EE*% was calculated by the following equation:(2)$$EE=\frac{Actual\;drug\;amount}{Theoretical\;drug\;amount}\times 100$$

The *Y* of the fabricated fibers was calculated by the following equation:(3)$$Y\%=\frac{Actual\;amount\;of\;fibres}{Theoretical\;amount\;of\;fibers}\times 100$$
where the theoretical amount of fibers is the amount of solid materials (polymer and drug) that was injected throughout the electrospinning system. The findings represent the mean (± SD) of at least three replicates.

### 2.9. Drug Release Profile

To determine the drug release, a defined weight of approximately 4 mg of the drug-loaded fibers was placed into 25 mL PBS (pH 7.4). The release study was carried out at 37 °C and 50 rpm in a thermostatically shaking incubator. Samples of 3 mL were withdrawn from the buffer solution after 5, 10, 15, 30, 60, 120, and 180 min, and equivalent volumes of pre-warmed fresh buffer were replaced to maintain sink condition. The amount of drug in the release buffer was determined by LAMBDA 850+ UV/Vis Spectrophotometer (PerkinElmer, Waltham, MA, USA) at 340 nm. The cumulative release percentages were measured as a function of time and were calculated based on the following equation:(4)$$\mathrm{Cumulative}\;\mathrm{amount}\;\mathrm{of}\;\mathrm{release}\;\left(\%\right)=\frac{\mathrm{Ct}}{C\infty }\times 100$$
where Ct is the amount of halicin released at time t and C∞ refers to the total amount of drug-loaded into the fibers. The findings represent the mean (±SD) of at least three replicates.

### 2.10. Bacterial Suspensions Preparation for Microbiological Testing

The American Type Culture Collection (ATCC) bacteria were used as controls. Multidrug-resistant (MDR) bacteria were used to compare their data with the controls. The gram-positive bacteria were presented by *Staphylococcus aureus* (*S. aureus*) ATCC 29213 and BAA 977 as an ATCC clinical isolate. While the gram-negative bacteria were shown by *Escherichia coli* (*E. coli*) ATCC 22925, and *Acinetobacter baumannii* (*A. baumannii*) ATCC BAA 747 as control and *A. baumannii* 3086 as an MDR clinical isolate. The suspensions (inoculums) of bacteria were prepared in Mueller–Hinton broth by measuring 0.5 McFarland. All bacterial species were cultured on Mueller–Hinton agar media and incubated overnight at 37 °C.

### 2.11. The Minimum Inhibitory Concentration (MIC)

To determine the MICs of halicin, bacterial inoculums of *E. coli* ATCC 22925, *S. aureus* BAA 977, *A. baumannii* ATCC BAA 747, and an MDR clinical isolate of *A. baumannii* 3086 were prepared and standardized. A serial dilution of the drug was prepared (256, 128, 64, 32, 16, 8, 4, 2, 1, 0.5, 0.25, 0.125 μg/mL) in Mueller–Hinton broth and added to 96-well plates, where one line of the plate contained no drug (i.e., bacteria only) as growth control. The bacterial suspensions were added to each well to achieve a final inoculum of 1 × 10^6^ colony-forming-unit/mL (CFU/mL). All 96-well plates were incubated at 37 °C overnight with continuous shaking speed at 120 rpm, following Kampshoff et al. [39]. MIC’s endpoint was measured at an absorbance of 600 nm using CytationTM 3 Cell Imaging Multi-Mode Reader (BioTek Instruments, Winooski, VT, USA). The turbidity of the solution (observed visually) was also an indication of the microbial growth, with the MIC being defined as the minimum concentration at which no growth was observed.

### 2.12. Antibacterial Zone of Inhibition Assay

To evaluate the antibacterial activity of the drug-loaded fibers, the zone of inhibition test on five bacterial strains was used: four ATCC standard strains, *E. coli* ATCC 22925, *S. aureus* ATCC 29213, *S. aureus* BAA 977, and *A. baumannii* ATCC BAA 747, and one clinical isolate of MDR *A. baumannii* 3086. The test was performed according to Karataş et al.’s [40] zone of inhibition study of ofloxacin/PCL fibers. The blank and drug-loaded fibers were tested against all five bacterial strains, cultured in Mueller–Hinton agar media, and incubated at 37 °C overnight. According to the MIC results, a certain weight of the drug-loaded fibers, which corresponded to the MIC against each bacterium, was measured. A final concentration of 1 × 10^6^ CFU/mL inoculum was equally distributed on the agar plates’ surface. The drug-loaded fibers and blank fibers in equivalent weights to the drug-loaded fibers were positioned on the agar plates’ center. Microbiological discs that contain equal concentrations to the halicin-loaded fibers were used as a test control. The diameters of the clear areas of ‘no growth’ were recorded in millimeters (mm). The results represent the mean (±SD) of at least three replicates.

### 2.13. Statistical Analysis

The mean, standard deviation (SD), regression equation, and correlation coefficient (r^2^) were all calculated using OriginPro 2016 software (OriginLab Corporation, Northampton, MA, USA).

## 3. Results & Discussion

Electrospun fibers’ utilization as medicated wound dressings is promising due to their reliable application on wound sites, prevention of high systemic drug concentrations, and increased patient compliance with comprehensive wound care [41,42,43]. An additional advantage is because of the use of a hydrophilic polymer, such as PVP in this present study, it is easier to clean off the remaining dressing from the wound surface. Therefore, PVP was used for the fabrication of halicin/PVP electrospun fibers as an antibacterial dressing.

### 3.1. Morphological Characteristics of the Electrospun Fibers

Drug-loaded electrospun fibers were successfully prepared using 8% (*w/v*) of PVP and 1% (*w/v*) of halicin, dissolved in ethanol, to obtain an 8:1 polymer: drug ratio. Halicin/PVP fibers demonstrated a smooth surface, lack of beads and pores, with an average diameter of 1.2 μm ± 0.3 μm, shown in Figure 1. Besides, no halicin crystals were observed on the surface of these fibers, indicating that this drug was successfully integrated within the fibers. Similar successful criteria of preparing drug-loaded PVP fibers were obtained in Maslakci et al. [44] and Baskakova et al. [45], who were able to load ibuprofen and acetylsalicylic acid, acyclovir, ciprofloxacin, and cyanocobalamin into PVP fibers but with nano-sized fiber diameters.

### 3.2. Fourier-Transform Infrared Spectroscopy (FTIR)

The FTIR spectroscopy was conducted to analyze any structural changes that might have occurred due to the incorporation of halicin into PVP fibers (i.e., the compatibility between the drug and the polymer) and to elucidate the drug–polymer molecular interactions. The chemical structure of halicin is shown in Figure 2. In contrast, the FTIR spectrum of the pure PVP polymer, halicin, and PM showed the characteristics peaks of each pure raw material, as illustrated in Figure 3. The peaks located at 1653, 1420, and 1269 cm^−1^ for PVP were assigned to the stretching vibrations of C=O, C-H (in an aliphatic compound), and C–N (in aromatic amine), respectively. A broad band was observed at 3418 cm^−1^, which indicated O-H stretching, owing to the hygroscopic nature of PVP. This spectrum is consistent with Tawfik et al. [11]. Since the stretching peak of PVP at 3418 cm^−1^ and 1653 cm^−1^ had strong bonding interactions, these peaks were also observed in the PM blends, the blank fibers, and the drug-loading fibers, as shown in Figure 3.

The FTIR spectrum for halicin demonstrated a stretching at the function group region representing 3366–3116 cm^−1^ (secondary amines N–H stretching), a spike-shaped band resembling a canine tooth. Very sharp two bands at 1520 cm^−1^ and 1653 cm^−1^, which are asymmetric nitro N-O and C=O stretching, respectively (Figure 3). Since nitro compounds (Figure 2), in general, have a strong bonding interaction, it was also observed on both PM blends and drug loading fibers to indicate the presence of the halicin (Figure 3). At the fingerprint region, the halicin spectrum bands between 761 and 734 cm^−1^, which represent the thiazole ring stretching (Figure 2 and Figure 3). The substitution of this ring at 692–668 cm^−1^ (i.e., organosulphur C-S stretching) was also observed as a characteristic peak at the drug loading and PM spectra, as an indication for the presence of the drug.

This result demonstrated halicin within the drug-loaded fibers and the PM with the absence of chemical interaction between the drug and the polymer (i.e., the lack of any changes in the structural integrity of the drug and polymer) that may arise due to the electrospinning process.

### 3.3. X-Ray Diffraction (XRD)

XRD diffraction patterns for the pure polymer and drug are presented in Figure 4. PVP is an amorphous polymer and therefore demonstrated a broad halo pattern consistent with Asawahame et al., [46] and Tawfik et al. [11]. Very distinctive reflections of halicin were observed at 7.00°, 14.52°, 16.80°, 17.66°, 19.86°, 20.30°, 20.64°, 22.50°, 22.90°, 25.40°, 28.14°, 28.42°, 29.46°, 30.34°, 31.62°, 32.16°, 32.84°, 35.02°, 38.66°, 43.12°, 43.78°, 45.40°, and 47.20°, indicating the crystallinity nature of this drug. The more intense diffraction peaks of the drug at 19.86°, 20.30°, 20.64°, 28.14°, 28.42°, 29.46°, 30.34°, 31.62°, 32.16°, 32.84°, and 35.02° were observed in the PM, while they were absent in the drug-loaded fibers that exhibited broad halo appearance similar to the PVP and blank fibers diffractograms, as shown in Figure 4. This result can confirm the molecular dispersion of the drug in the drug-loaded fibers and the lack of any crystalline structure in these fibers. This finding is in agreement with the results found in the drug-loaded PVP fibers of Illangakoon et al. [47], Wang et al. [48], and Asawahame et al. [46]. In addition, an SEM image of the pure drug confirmed its crystallinity, as illustrated in Figure 5. However, more investigation is required to identify the exact crystal lattice of the drug.

### 3.4. UV Standard Curve

The UV assays were simple and rapid, with the drug being observed in DW and PBS (pH 7.4) for the determination of the DL and release of the drug-loading fibers, respectively. The drug calibration curves in DW and PBS showed good linearity with regression equation (and r^2^) of y = 0.0541x + 0.0022 (0.9994) and y = 0.0637x + 0.0039 (0.9995), respectively, as illustrated in Figure 6.

### 3.5. Drug Loading (DL), Encapsulation Efficiency (EE%), and Fiber Yield (Y)

The DL and EE% for the drug-loading fibers were measured as 66.3 μg/mL and 60%, respectively. This EE% was lower than the previous studies of Dubey et al. [49] on Cyclosporine A/PVP fibers, Kamble et al. [50] on Irbesartan (IBS)/PVP fibers, Wang et al. [51] on Janus zein/PVP and Rramaswamy et al. [52] on tetrahydro curcumin (THC)/PVP fibers, which will require further investigation. The Y of the drug-loaded fibers was estimated as 82 ± 17%, and the remaining amount was lost probably during the processing (i.e., around the collector) and on the aluminum foil (i.e., remained attached to the foil after peeling off the fibers).

### 3.6. Drug Release Profile

The release of halicin-loaded fibers exhibited a burst release of approximately 60% of the drug after 15 min, owing to the polymer’s hydrophilic nature (i.e., PVP), which would accelerate the fibrous-mat disintegration and finally, the releasing of the loaded drug, as shown in Figure 7. A full drug release was obtained after 120 min.

This accelerated release profile was expected due to the polymer’s hydrophilic nature (PVP) that will disintegrate fast upon contact with the buffer (PBS). This finding is consistent with Li et al. [53], Sriyanti et al. [54], and Moydeen et al.’s [55] studies on carvedilol-, α-mangostin-, and ciprofloxacin-loaded PVP fibers, respectively. The in vitro release study’s trend is summarized as follows: diffusion-dependent drug release occurred at the initial time points (5–15 min) leading to burst release, followed by polymer degradation, which accounted for the full drug release (at about 30 to 120 min).

### 3.7. Antibacterial Activity

Halicin demonstrated antibacterial effectiveness against different gram-positive and gram-negative bacterial strains such as *S. aureus, A. baumannii*, and *E. coli*, which was a concentration-dependent effect. The MIC of halicin was measured in a concentration ranging from 0.125 to 256 µg/mL. As shown in Figure 8 and Figure 9, the MIC of halicin was determined to be 16 μg/mL and 32 μg/mL against *S. aureus* and *E. coli*, respectively, while higher concentrations of 128 μg/mL and 256 μg/mL were reported against *A. baumannii* and MDR *A. baumannii*, respectively. Accordingly, the drug-loaded fibers were tested at drug concentrations of 128 and 265 μg/mL to ensure a good effect against all tested bacterial strains.

The antibacterial activity of the drug-loaded fibers was evaluated by the zone of inhibition test (also known as the disc diffusion test) against the similar bacterial strains that were used in the MIC study, in addition to another *S. aureus* strain (ATCC 29213), since this bacterium is commonly found in pressure ulcer patients. Well-defined inhibition zones for the drug-loaded fibers and halicin discs were observed with variable diameters (Figure 10). The zone of inhibition diameter was around 21 mm against *S. aureus* (ATCC 29213), 30 mm against *S. aureus* (BAA 977), 27 mm against *E. coli*, 20 mm against *A. baumannii,* and 18 mm against MDR *A. baumannii* (Table 1). Moreover, the zone of inhibition for halicin discs, in equivalent concentrations to the drug-loaded fibers, was approximately 28 mm against both *S. aureus* strains, 20 mm against *E. coli*, 24 mm against *A. baumannii,* and 21 mm against MDR *A. baumannii* (Table 1). These variant results between the drug-loaded fibers and discs might be due to the variation in the drug loading that can occur within the formulation upon electrospinning (since it is not an absolute fixed dose) or owing to processing errors, such as inaccurate weighing of the drug-loaded fibers that would lead to slightly different zones of inhibition. However, the important finding of this study is that halicin fibers were successfully prepared by electrospinning, and the drug retained its antibacterial activity against different bacterial strains that commonly cause a pressure ulcer.

Stokes et al. [31] reported the antibacterial effect of halicin ointment (in a concentration of 0.5% *w/v*) against several bacterial strains, including *A. baumannii* ATCC 17978 and *Clostridium difficile 630*. Additionally, *E. coli* BW25113 did not develop any resistance against halicin over the 30-day treatment period due to the drug’s ability to inhibit this bacterium by suppressing their electrochemical gradient through their cell membranes that are required to generate ATP. This study also suggested that it is less likely that the drug would develop bacterial resistance against different *E.coli* strains due to this unique mechanism of action. Furthermore, other studies demonstrated electrospinning’s ability to fabricate antibacterial dressing that retained their antibiotic effectiveness successfully. PVP nanofibers loaded with trimethoprim showed an inhibitory effect against both *S. aureus* (ATCC 33591) and *E.coli* (ATCC 35218) [56]. Maslakci et al. [44] demonstrated the antibacterial activity of PVP-loaded ibuprofen or acetylsalicylic acid against *S. aureus* (ATCC 25923) and *Bacillus subtilis*. El-Newehy et al. [57] evaluated the antibacterial activity of metronidazole-loaded into a PVP/PEO blend, which was active against *E. coli* and *P. aeruginosa*.

## 4. Conclusions

Pressure ulcer is a very common skin infection that affect patients who are admitted to the hospital, and it requires special attention in order to prevent the development of any further infections that may be life-threatening to these patients. Various bacterial strains are known to cause skin infections. However, resistance of these bacteria is prevalent, which has encouraged the discovery of new antibiotics to overcome such microorganisms. Halicin was found to inhibit several bacterial strains, including some resistant strains, owing to its unique antibacterial mechanism. In this study, halicin fibers were developed as an antibacterial dressing that can be used for patients with pressure ulcers against several bacteria, in particular, MDR *A. baumannii,* which is known for its difficulty of treatment. Halicin fibers were successfully prepared by electrospinning with un-beaded, non-porous, and smooth fibers’ surface and a diameter of 1.2 μm (±0.3 μm). FTIR and XRD were used to evaluate this novel drug’s physical characteristics, and the findings demonstrated the crystalline nature of the drug while it was molecularly dispersed upon electrospinning. The drug-loaded fibers were able to release the drug while retaining its antibacterial efficacy against different bacterial strains, including the MDR *A. baumannii*.

In conclusion, this study showed that halicin/PVP electrospun fibers are a potential antibacterial dressing that can be topically applied to infected skin, particularly in a pressure ulcer. However, more in vitro and in vivo studies of this system and the drug are required to evaluate their safety and efficacy more comprehensively.

## Figures and Tables

**Figure 1 pharmaceutics-12-01189-f001:**
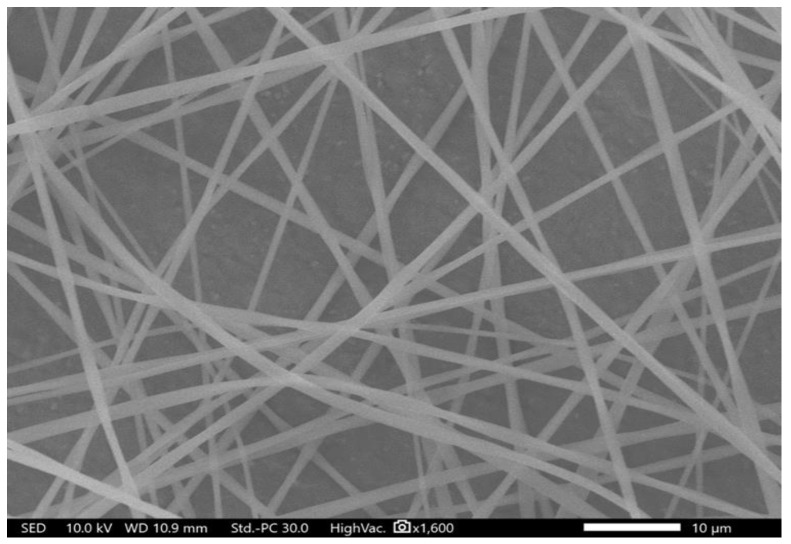
Drug-loaded electrospun fibers showing a smooth, un-beaded, and un-porous appearance of the yielded fibers indicating their successful preparation.

**Figure 2 pharmaceutics-12-01189-f002:**
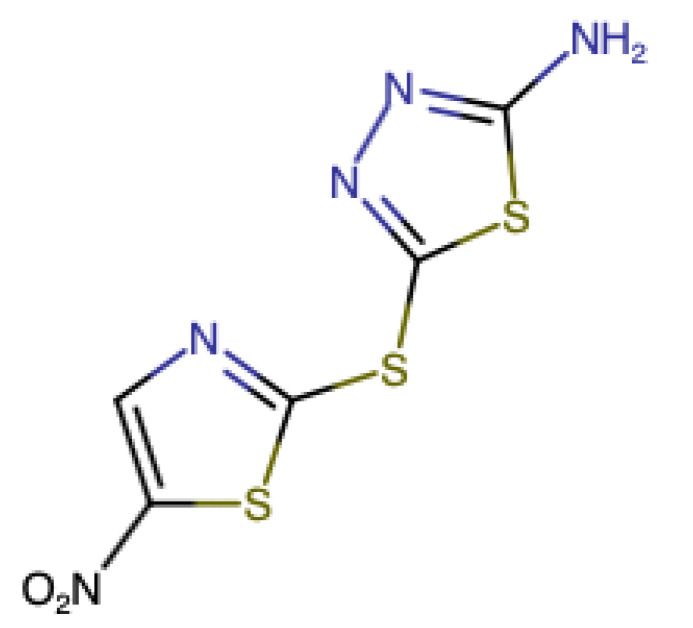
The chemical structure of halicin, which was drawn by chem-space.com.

**Figure 3 pharmaceutics-12-01189-f003:**
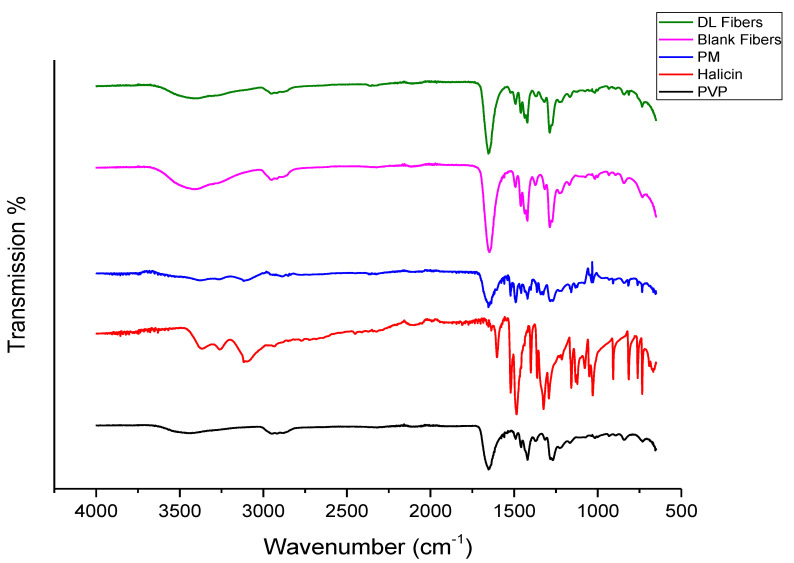
Fourier-transform infrared spectroscopy (FTIR) transmissions of polyvinylpyrrolidone (PVP), halicin, physical mixture (PM), blank and drug-loaded fibers, showing the distinctive drug peaks at 761 to 734 cm^−1^ that appear in the PM and the drug-loaded fibers compared to the blank fibers.

**Figure 4 pharmaceutics-12-01189-f004:**
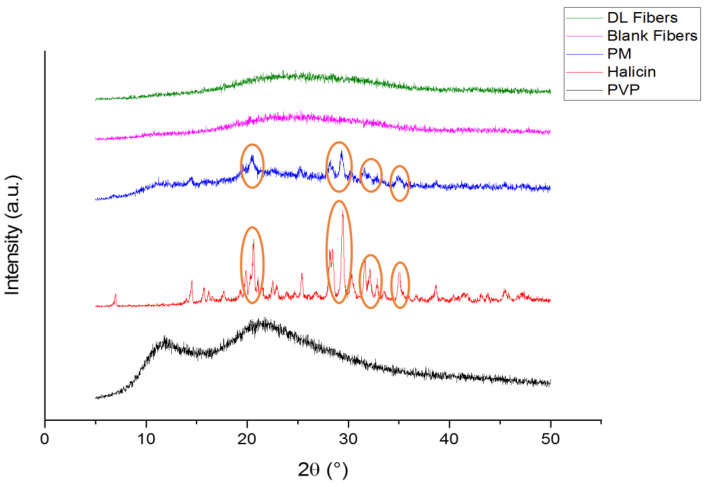
X-ray diffraction (XRD) patterns of PVP, halicin, PM, blank and drug-loaded fibers showing that the drug is in the crystalline form (presence of characteristic peaks) while the polymer is in the amorphous form (broad halos). The presence of halicin distinct peaks is also present in the PM which are absent in the drug-loaded fibers indicating the molecular dispersion of the drug within the fibers.

**Figure 5 pharmaceutics-12-01189-f005:**
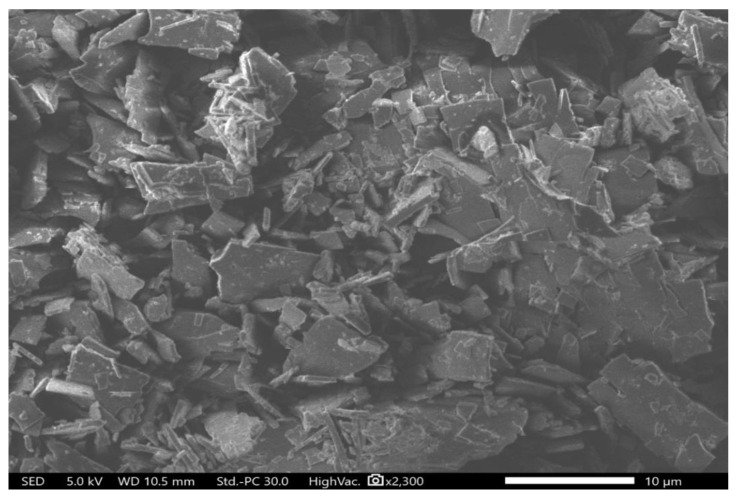
Scanning electron microscopy (SEM) image of halicin crystal showing the drug crystal structure.

**Figure 6 pharmaceutics-12-01189-f006:**
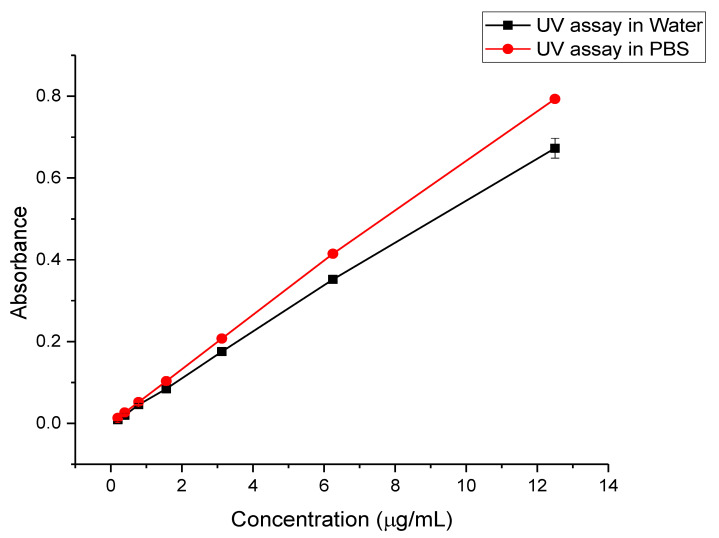
Halicin UV calibration curves using deionized water (DW) (for drug loading (DL) determination) and phosphate buffer saline (PBS) pH 7.4 (for the release study). Both curves showed good linearity at a concentration range of 12.5 to 0.2 μg/mL, with the r^2^ of 0.9994 and 0.9995 for DW and PBS, respectively.

**Figure 7 pharmaceutics-12-01189-f007:**
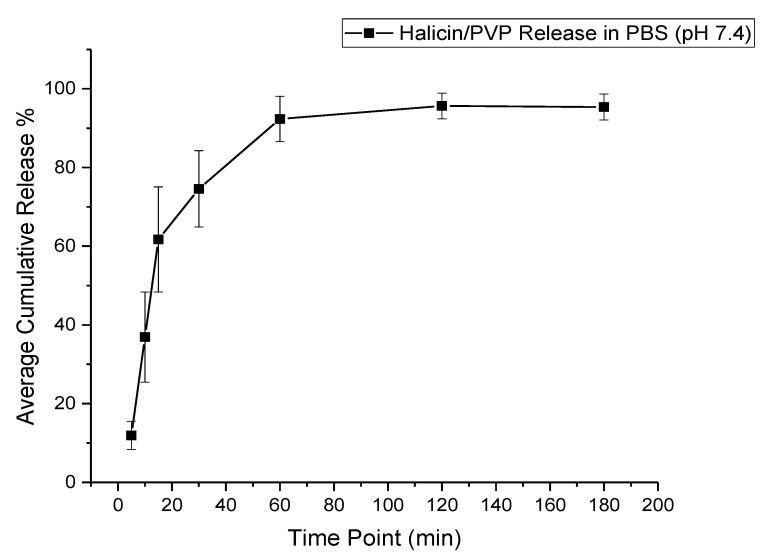
Cumulative release profile of the drug-loaded fibers showing a burst release of the drug (~60%) after 15 min and a full drug release after 120 min.

**Figure 8 pharmaceutics-12-01189-f008:**
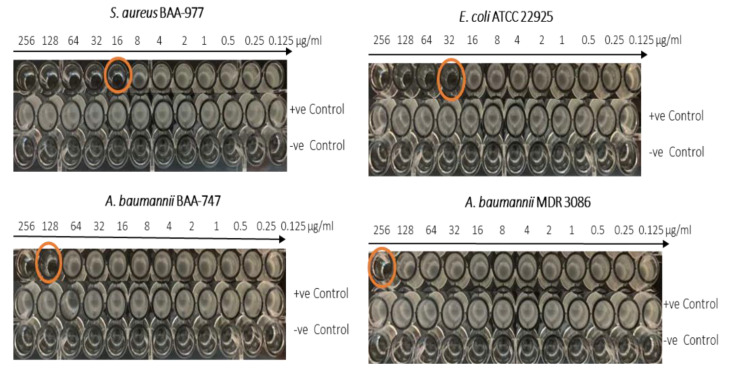
The minimum inhibitory concentration (MIC) values of halicin against *S. aureus* (BAA 977), *E. coli* (ATCC 22925), *A. baumannii* (BAA 747), and *A. baumannii* (MDR 3086), in which the minimum concentration (orange circled) is defined as the lowest concentration of no bacterium growth. The MIC was found at 16 μg/mL, 32 μg/mL, 128 μg/mL, and 256 μg/mL against *S. aureus*, *E. coli*, *A. baumannii*, and MDR *A. baumannii*, respectively.

**Figure 9 pharmaceutics-12-01189-f009:**
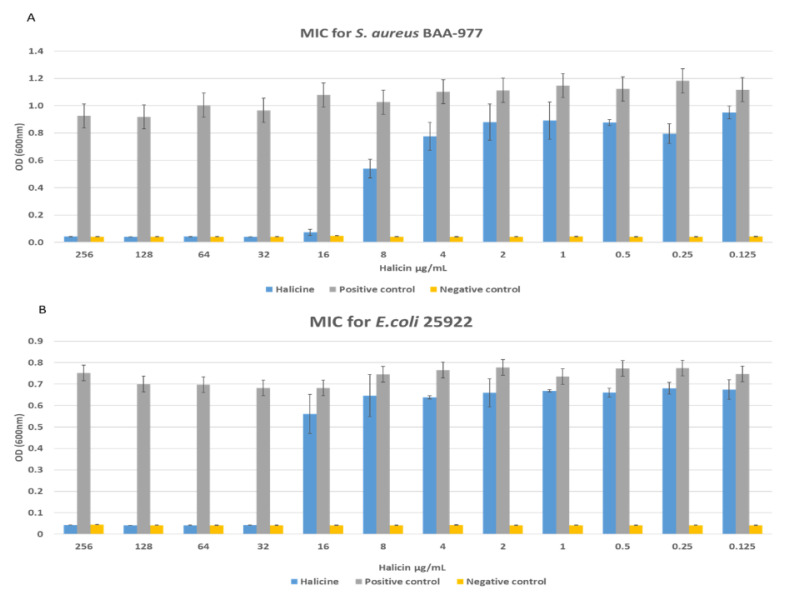
The MIC values were found at 16 μg/mL, 32 μg/mL, 128 μg/mL, and 256 μg/mL against *S. aureus* (**A**), *E. coli* (**B**), *A. baumannii* (**C**), and MDR *A. baumannii* (**D**), respectively.

**Figure 10 pharmaceutics-12-01189-f010:**
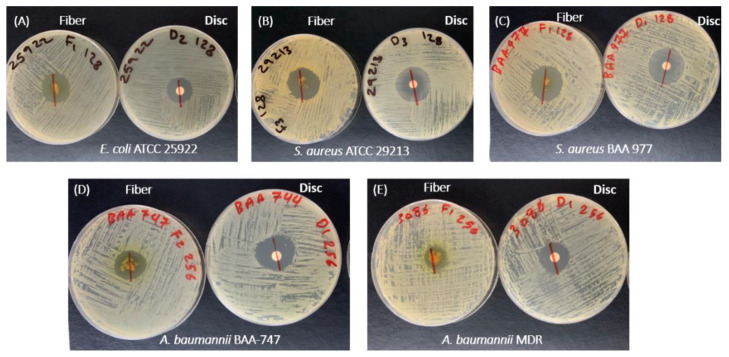
Zone of inhibition of the drug-loaded electrospun fibers compared to an equivalent concentration of the free drug in a sterile disc. The zone of inhibition of the drug-loaded fibers was 27 mm, 21 mm, 30 mm, 20 mm, and 18 mm for *E. coli* (**A**), *S. aureus* ATCC 29,213 (**B**), *S. aureus* BAA 977 (**C**), *A. baumannii* BAA-747 (**D**), and MDR *A. baumannii* (**E**), respectively, while it was 20 mm, 28 mm, 28 mm, 24 mm, and 21 mm for the halicin discs, respectively.

**Table 1 pharmaceutics-12-01189-t001:** The zone of inhibition diameters of halicin discs, blank and drug-loaded fibers against *E. coli* (ATCC 22925), *S. aureus* (ATCC 29213), *S. aureus* (ATCC BAA 977), *A. baumannii* (BAA 747), and *A. baumannii* (MDR 3086). Both the drug-loaded fibers and halicin discs showed clear zones of inhibitions against all bacterial strains, while the blank fibers showed no effect against those bacteria. The results represent the mean (±SD) of *n* = 3.

Bacterial Strain	Zones of Inhibition (mm)		
	Halicin Disc	Blank Fibers	Drug-Loaded Fibers
*E. coli* (ATCC 22925)	20 ± 0	0	27 ± 3
*S. aureus* (ATCC 29213)	28 ± 1	0	21 ± 1
*S. aureus* (ATCC BAA 977)	29 ± 1	0	30 ± 2
*A. baumannii* (BAA 747)	25 ± 1	0	21 ± 2
*A. baumannii* (MDR 3086)	22 ± 1	0	18 ± 1

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
