# Peer review of "The Delivery of the Novel Drug ‘Halicin’ Using Electrospun Fibers for the Treatment of Pressure Ulcer against Pathogenic Bacteria"

_pharmaceutics, 2020, doi:10.3390/pharmaceutics12121189_

Round 1
Reviewer 1 Report
This is an interesting work about the delivery of the novel drug ‘Halicin’ using electrospun fibers for the treatment of pressure ulcer against pathogenic bacteria. The science is solid and the characterization is complete. Overall, I think it would qualify for Pharmaceutics. However, I still have some minor concerns before I can recommend it for publication.
(1) Some images are of low resolution, which need to be corrected. For example, Figure 3, Figure 8.
(2) The error analysis of this work is not properly handled in some places. The data presented have too many significant figures for standard deviation. For example, in table 1. "27.33 ± 3.06" should be "27±3".
(3) In the introduction section, the authors mainly introduce Halicin for the treatment of pressure ulcer against pathogenic bacteria. However, the authors do not introduce the application of electrospun nanofibers, the other part of the materials. Actually electrospun nanofibers have been widely used for biomedical application (Journal of Materials Chemistry B, 2019, 7, 709–729), drug delivery (ACS Applied Materials & Interfaces, 2019, 11, 12880-12889; Carbohydrate Polymers, 2016, 151 1240–1244; Carbohydrate Polymers, 2017, 169, 198-205) as well as air filtration (Macromolecular Materials and Engineering, 2017, 302(1), 1600353-1600380; Macromolecular Materials and Engineering, 2018, 1800336-1800354). I would like to suggest the authors include those emerging applications of electrospun nanofibers in the introduction section.
Based on the above concerns, I would suggest a minor revision.
Author Response
We thank the referees for their thoughtful and helpful comments. We respond to each point below; their suggestions have undoubtedly improved the paper, for which we are very grateful.
Reviewer #1:
This is an interesting work about the delivery of the novel drug ‘Halicin’ using electrospun fibers for the treatment of pressure ulcer against pathogenic bacteria. The science is solid and the characterization is complete. Overall, I think it would qualify for Pharmaceutics. However, I still have some minor concerns before I can recommend it for publication.
- Some images are of low resolution, which need to be corrected. For example, Figure 3, Figure 8.
The resolution of all Figures has been enhanced.
- The error analysis of this work is not properly handled in some places. The data presented have too many significant figures for standard deviation. For example, in table 1. "27.33 ± 3.06" should be "27±3".
This has been solved now.
- In the introduction section, the authors mainly introduce Halicin for the treatment of pressure ulcer against pathogenic bacteria. However, the authors do not introduce the application of electrospun nanofibers, the other part of the materials. Actually electrospun nanofibers have been widely used for biomedical application (Journal of Materials Chemistry B, 2019, 7, 709–729), drug delivery (ACS Applied Materials & Interfaces, 2019, 11, 12880-12889; Carbohydrate Polymers, 2016, 151 1240–1244; Carbohydrate Polymers, 2017, 169, 198-205) as well as air filtration (Macromolecular Materials and Engineering, 2017, 302(1), 1600353-1600380; Macromolecular Materials and Engineering, 2018, 1800336-1800354). I would like to suggest the authors include those emerging applications of electrospun nanofibers in the introduction section.
This paragraph has been added “Electrospinning is a versatile process for fabricating fibers with diameters that can reach to the nanoscale. Owing to the large surface area, ease of preparation, flexibility and tensile strength of these fibers, they have been utilized in a wide range of applications, including antimicrobial dressings tissue engineering, wound healing, drug delivery and in the manufacture of textiles, cosmetics, molecular filtration, fuel cells and sensor systems [10,11]. Besides, the ability to encapsulate different active compound, such as drugs, peptides, proteins, nucleic acids and cells, makes these fibers attractive drug delivery platforms. The diversity of the natural and synthetic occurring polymers allows the fabrication of fibers that are compatible with the biological tissues, which can hold a potential significance in tissue regeneration applications. The resemblance of the fibers to the elements of tissue extracellular matrix (ECM), which plays an essential role in wound healing, allows the use of these fibers as scaffolds to accelerate wound closure [11]. ”
We didn’t want to get out-of-scope, so we focused on the biological application of the fibers more than the industrial application.
Reviewer 2 Report
Dear Authors,
in your interesting manuscript, the following points should be added/changed to further improve it:
- Names: The last author needs only one "*", not two.
- Graphical abstract: Please use identical fonts throughout this image. Please enlarge the axis labels of the graph.
- Please put the refences in square brackets.
- "bacterial resistant" - you mean "antibiotics resistant bacteria" or "multi-resistant bacteria"
- lines 90 ff: please don't write the first letters in each chemical capitalized
- line 90: please add (PVP) behind Polyvinylpyrrolidone to define the abbreviation
- lines 97, 99: Why were these concentrations chosen?
- line 102: What do you mean with "monoaxial" or "single-layered" fibers? That the fibers are oriented? Or that they don't have a core-shell structure?
- Eq. 1: What does "amount" mean, are these values masses or volumes or something else?
- line 148: 4.1 mg +- 0.1 mg
- Fig. 3: Please switch the numbers on the x-axis from large to small numbers. Please superscript the "minus" sign in cm^-1. Are the lines vertically offset for clarity? If so, please mention this in the caption (ditto in Fig. 4).
- Fig. 6: What does the x-axis unit mean?
- How do the values in Fig. 6, probably given in ug/mL, correspond with the percentages in Fig. 7?
- Conclusion: The first two sentences sound too near to Abstract and Introduction, please think about skipping them.
Author Response
We thank the referees for their thoughtful and helpful comments. We respond to each point below; their suggestions have undoubtedly improved the paper, for which we are very grateful.
Reviewer #2:
Dear Authors,
in your interesting manuscript, the following points should be added/changed to further improve it:
- Names: The last author needs only one "*", not two.
The star has been changed to a and b for the co-corresponding and corresponding authors, respectively.
- Graphical abstract: Please use identical fonts throughout this image. Please enlarge the axis labels of the graph.
This has been changed as recommended.
- Please put the refences in square brackets.
This has been changed as recommended.
- "bacterial resistant" - you mean "antibiotics resistant bacteria" or "multi-resistant bacteria"
“antibiotic-resistant” has been used instead.
- lines 90 ff: please don't write the first letters in each chemical capitalized
This has been changed as recommended.
- line 90: please add (PVP) behind Polyvinylpyrrolidone to define the abbreviation
This has been changed as recommended.
- lines 97, 99: Why were these concentrations chosen?
PVP in a concentration of 8%
The polymer concentration was chosen in accordance to the optimization processing that took place prior to this work, which involves different concentrations (6, 8, and 10% w/v).
halicin at a concentration of 1% (w/v)
This drug concentration was chosen to achieve a polymer:drug ratio of 8:1, which is a recommended ratio that ensure a complete encapsulation of the drug within the fibrous matrix. This sentence has been added in the manuscript.
- line 102: What do you mean with "monoaxial" or "single-layered" fibers? That the fibers are oriented? Or that they don't have a core-shell structure?
‘Monoaxial fiber’ is a term that describes a one-layered (i.e. single –layered) fiber that is actually not core/shell (i.e. coaxial fiber). It is a very common term in electrospinning and a description was followed for those who are not expert with this technique.
- Eq. 1: What does "amount" mean, are these values masses or volumes or something else?
The drug mass that was calculated based on the calibration curve (concentration vs absorbance)
- line 148: 4.1 mg +- 0.1 mg
This has been changed to 4 mg, as the word approximately is there already.
- Fig. 3: Please switch the numbers on the x-axis from large to small numbers. Please superscript the "minus" sign in cm^-1. Are the lines vertically offset for clarity? If so, please mention this in the caption (ditto in Fig. 4).
This has been edited
- Fig. 6: What does the x-axis unit mean?
mcg/mL has been changed to μg/mL (microgram/mL)
- How do the values in Fig. 6, probably given in ug/mL, correspond with the percentages in Fig. 7?
Each sample from the required time point (e.g. 5 min time point) was scanned for absorbance in the UV spec. Then the concentration correlated to this absorbance was identified by the developed calibration curve (in PBS pH 7.4). This concentration was changed to drug amount by multiplying by the volume used. This also was the case in measuring the DL but we used the calibration curve of the drug dissolved in deionized water (DW).
- Conclusion: The first two sentences sound too near to Abstract and Introduction, please think about skipping them.
These sentences have been added instead “Pressure ulcer is a very common skin infection that affect patients who are admitted to the hospital and it requires a special attention, in order to prevent the development of any further infections that may be life-threatening to these patients. Various bacterial strains are known to cause skin infections. However, resistance of these bacteria is prevalent, which has encouraged the discovery of new antibiotics to overcome such microorganisms.”